# Odorant Receptors Expressing and Antennal Lobes Architecture Are Linked to Caste Dimorphism in Asian Honeybee, *Apis cerana* (Hymenoptera: Apidae)

**DOI:** 10.3390/ijms25073934

**Published:** 2024-04-01

**Authors:** Haoqin Ke, Yu Chen, Baoyi Zhang, Shiwen Duan, Xiaomei Ma, Bingzhong Ren, Yinliang Wang

**Affiliations:** 1Jilin Provincial Key Laboratory of Animal Resource Conservation and Utilization, School of Life Science, Northeast Normal University, Changchun 130024, China; kehq907@nenu.edu.cn (H.K.); chenyu810@nenu.edu.cn (Y.C.); zhangbaoyi@nenu.edu.cn (B.Z.); duansw693@nenu.edu.cn (S.D.); maxm769@nenu.edu.cn (X.M.); bzren@nenu.edu.cn (B.R.); 2Key Laboratory of Vegetation Ecology, MOE, Northeast Normal University, Changchun 130024, China

**Keywords:** apis olfactory system, antennal lobes, bee glomeruli, caste dimorphism, odorant receptor

## Abstract

Insects heavily rely on the olfactory system for food, mating, and predator evasion. However, the caste-related olfactory differences in *Apis cerana*, a eusocial insect, remain unclear. To explore the peripheral and primary center of the olfactory system link to the caste dimorphism in *A. cerana*, transcriptome and immunohistochemistry studies on the odorant receptors (ORs) and architecture of antennal lobes (ALs) were performed on different castes. Through transcriptomesis, we found more olfactory receptor genes in queens and workers than in drones, which were further validated by RT-qPCR, indicating caste dimorphism. Meanwhile, ALs structure, including volume, surface area, and the number of glomeruli, demonstrated a close association with caste dimorphism. Particularly, drones had more macroglomeruli possibly for pheromone recognition. Interestingly, we found that the number of ORs and glomeruli ratio was nearly 1:1. Also, the ORs expression distribution pattern was very similar to the distribution of glomeruli volume. Our results suggest the existence of concurrent plasticity in both the peripheral olfactory system and ALs among different castes of *A. cerana*, highlighting the role of the olfactory system in labor division in insects.

## 1. Introduction

In eusocial insects, different levels of chemical communications among individuals determine the labor division influencing their social behavior. For instance, in honeybees, each caste behaves as a specialized individual. Queens participate in hive breeding by using the volatile olfactory cues produced by drones to find suitable mates [1]. Drones are born for mating and after sexual maturity are attracted by queen pheromones to mate with the queen during nuptial flight [2]. Workers perform various responsibilities: the younger workers nurse the larvae and queen and perform hive cleaning under the influence of pheromones from brood and queen [3]; the older workers focus on forage, nectar collection, community defense, etc., and are more sensitive to flower scents and cuticular hydrocarbons [4,5]. The differential chemical signals are the basis of caste dimorphism and complex social cooperation in insects involving the olfactory sensing system [6,7].

In eusocial insects, caste dimorphism of the olfactory system is not only reflected at the morphological level but also the peripheral and central nervous system (CNS) levels [8]. At the peripheral level, workers have significantly higher numbers of odorant receptors (ORs) due to a specific 9-exon ORs clade, which is highly expressed on the ventral side of the worker’s antenna [9,10,11]; this might improve the perception of workers towards social signals. At the CNS level, drones have an evolved visual system with a larger optic lobe (OL), while workers have enhanced olfactory systems with larger antennal lobes (ALs) [12,13,14]. Moreover, the number, size, and relative positions of glomeruli have been associated with insect castes. For instance, some drones have several specific macroglomeruli (MG), a type of enlarged glomeruli that enhances pheromone sensing [15].

*Apis mellifera* and *A. cerana* are two closely related species with many similarities in morphology and social behavior, but they differ in some aspects such as resistance, pollination ability, etc. [16]. For instance, *A. cerana* has a stronger resistance to some pesticides, extreme temperatures, parasites, and pathogens [17,18]. *A. cerana* is also capable of collecting larger pollen from a wide range of flowers under various habitats [19,20]. Such differences between *A. mellifera* and *A. cerana* may originate from the differentiation of the olfactory system. *A. mellifera* is a model species and therefore has been extensively studied for ORs and ALs [3,9]. However, there is a lack of information on the caste-specific ORs expression profile in *A. cerana*, especially regarding the full map of ALs.

*A. cerana*, with a wide range of host plants and high pollination efficiency, however, has an olfactory system that is poorly understood compared with its close relative *A. mellifera*. The colonies of *A. cerana* are on a significant decline due to the massive use of pesticides and pathogens invasion, especially with the introduction of *A. mellifera* in commercial apiculture [21,22]. In the past century, the number of *A. cerana* has decreased by 80%, and their distribution area has reduced by 75% in China [23]. In this study, we used RNA-seq and immunohistochemical techniques to explore the caste dimorphism of the olfactory system in *A. cerana*, which would improve our understanding of the social behavior and olfaction differentiation in *A. cerana*. This information will potentially benefit the management, feeding, and conservation of this native pollinator.

## 2. Results

### 2.1. Overview of Transcriptome

After removing adapter sequences and trimming low-quality nucleotides, 21 libraries were generated with clean reads size from 19.48 to 27.89 Mb. The GC content of clean data ranged from 34.70% to 41.54%, and the Q30 bases in all samples were more than 92%, indicating the reliability of RNA sequencing data. The mapping rate of samples against the reference genome ranged from 74.72% to 92.97% (Appendix A). In total, 307,615 transcripts were obtained, and 83,632 unigenes were annotated (Appendix A). Among different castes, the average gene expression levels in the antenna of queens were significantly higher than that of drones and workers (F = 16.181, df = 2, *p* < 0.05, Figure 1, Appendix A). All the transcriptome data were uploaded to the NCBI SRA (PRJNA932856).

### 2.2. Differential Gene Analysis in Antenna

The genes expressed in the antenna, the main olfactory organ, were selected for DEGs analysis among queens (QA), drones (DA), and workers (WA). The number of DEGs in the three groups was significantly different: QA vs. DA, 3252; WA vs. DA, 568; WA vs. QA, 3866. However, more genes were upregulated in QA vs. DA (1594) and WA vs. QA (2100) groups than in the WA vs. DA group (317) (Table 1). The GO enrichment analysis of upregulated genes showed that in the “molecular function” category, more DEGs in “odorant binding” were from queens and workers than that from drones, suggesting relatively low expression of such genes in drones than in queens and workers (Figure 2A,D). However, the “odorant binding” were ranked 529, respectively, in the WA vs. QA group and not shown in Figure 2G (Appendix A). Interestingly, workers exhibited much more upregulation of genes related to the development of the nervous system, such as the formation of synapses and the transmission of neurotransmitters, e.g., “nervous system development”, “neurogenesis”, and “generation of neurons” were the main “biological process”, while “synapse” values were the main “cellular component” (Figure 2H,I).

### 2.3. Identification and Homology Analysis of AcerORs

In total, 106 candidate AcerORs were identified from the transcriptome data of *A. cerana*. The length of these AcerORs was 336 to 478 amino acids, and 99 AcerORs were complete ORFs. Blast results showed that homology similarity between these AcerORs and those from other Hymenoptera species ranged from 25.13% to 100%. The best hit of most AcerORs were the orthologues ORs from *A. mellifera*, while some of the AcerORs (such as AcerOR34, AcerOR85b, and AcerOR91b, etc.) were best hit to the ORs in *Apis dorsata*, *Apis laboriosa*, and *Apis florea*. In addition, some AcerORs (AcerOR97, AcerOR155, and AcerOR159, etc.) were also found to be most similar to the ORs of *Bombus vancouverensis nearcticus*, *Bombus terrestris*, and *Temnothorax curvispinosus* (Appendix A). These results indicated that the ORs sequences were relatively conserved among tested Hymenoptera species.

An ML phylogenetic tree of ORs was constructed using the datasets from *A. cerana* (106 ORs), *A. mellifera* (170 ORs), and *A. florea* (180 ORs). The AcerORs were named by reference to their homology genes from AmelORs. Consequently, all ORs were divided into five distinct clades: clade I contains 18 AcerORs and in this clade, most AmelORs and AfloORs contain nine exons. Clade II consists of 6 AcerORs: Orco clade belonging to this clade is highly conserved among different species, with more than 99% identity in the species examined in this study. Clade III and clade IV contain similar AcerOR numbers (18 and 20, respectively), while clade V contains the largest number of AcerORs (44) (Figure 3).

### 2.4. Expression Profiles of AcerORs

We found that more *AcerORs* were specifically expressed in the antenna and were more in queens and workers than in drones (Table 2). Among the *AcerORs*, some did not show significant expression differences among castes, such as *AcerOR86b* and *AcerOR169a* (cluster II). In contrast, some *AcerORs* exhibited caste-specific expression, such as *AcerOR170* and *AcerOR22* in drones (cluster I); *AcerOR166b* and *AcerOR173* in queens (cluster IV); and *AcerOR75* and *AcerOR25b* in workers (cluster V). Meanwhile, a few *AcerORs* were highly expressed in other non-olfactory tissues, for example, *AcerOR96a*, *AcerOR96b*, *AcerOR141a*, and *AcerOR72* were highly expressed in the abdomen of drones, and *AcerOR1* was highly expressed in the thorax of queens and drones (cluster III) (Figure 4).

According to DEGs analysis, there were 33 candidate *AcerORs* with caste-specific differential expression, which were verified by RT-qPCR. As expected, the expressing patterns of most *AcerORs* were consistent with their FPKM values; only 7 *AcerORs* showed different results. RT-qPCR results showed that the expression of *AcerOR28b, AcerOR54b*, *AcerOR79b*, *AcerOR121*, and *AcerOR149b* was queen-biased, while *AcerOR96a* was expressed higher in drones. Meanwhile, *AcerOR149a* expression was nearly the same between queens and workers (Figure 5).

### 2.5. Caste-Specific Architecture of the Whole Brain and Antennal Lobes

The brain structure of *A. cerana* is very similar to other bees and consists of supraesophageal ganglia (SPG) and subesophageal ganglia (SOG). SPG is composed of the protocerebrum (PR) and deutocerebrum (DE), and PR is mainly composed of MB and the optic lobe (OL). MB consists of the calyx (mCa: medial calyx; lCa: lateral calyx), pedunculus (PED), and lobe (vL: vertical lobe, etc.). The OL of *A. cerana* is made up of the lobula (Lo), medulla (Me), and lamina (La). The ALs are an important part of DE that are paired in the posterior side of PR while connecting to the antenna (Figure 6).

We found that although the shape of ALs varied among castes, the distribution of glomeruli was quite similar. The ALs of queens and workers were almost round in shape but those of drones exhibited an oval shape. Queens and workers have several small glomeruli at the entrance of the antennal nerve to the Als, but the same was replaced by several larger glomeruli in drones. In all three castes, glomeruli were scattered at the edge of ALs, forming a hollow central fiber core without being synaptic. In general, there were more ventral side glomeruli than on the dorsal side. Most glomeruli were nearly spherical and ellipsoidal, but a few glomeruli were irregular in shape (Figure 7 and Appendix A).

### 2.6. 3D Model and Characteristics of Glomeruli

The 3D modelling results showed that the number of glomeruli increases from the posterior to the anterior side. A few large glomeruli were observed in the middle of the outermost portion of the ventral side, but no glomeruli were found on the opposite side. All the glomeruli formed a hollow sphere, and more glomeruli were distributed in the equatorial axis than the pole sides located in the outermost portion of the sphere. Moreover, there were several MGs in drones that were located at the edge of ALs (Figure 8 and Appendix A).

The number, total surface area, and total volume of ALs were calculated after three-dimensional reconstruction to find possible characteristic differences among castes. The number of glomeruli was almost the same in queens and workers; however, it decreased in drones. Queens and drones showed no significant difference in the total volume and surface area; however, both of these indices were significantly large in workers (Table 3).

The type of glomeruli was highly caste-specific in drones: VC, AV, PV, LD, and MD (ventral and dorsal side) were absent, and the number of AL, AM, PL, and PM (lateral and medial side) glomeruli were decreased. The types of glomeruli between queens and workers were quite similar, except for LD and MD (dorsal side), which were reduced in workers (Table 3). Notably, the three macroglomeruli (MGa, MGb, and MGd) were only detected in drones; MGb is the largest and occupies 12.59% of the total volume of glomeruli, MGa and MGd are comparatively smaller, with a total relative volume of about 7.5%. According to the position and shape, all glomeruli were clustered into classes 1–4. Among the three castes, the class 1 glomeruli were the maximum (more than 40%), followed by class 2 (about 30%). In workers, the number of class 3 and 4 glomeruli was nearly the same; however, there were more class 4 glomeruli in queens and drones (Appendix A).

### 2.7. Correlation between Glomeruli and ORs in A. cerana

The correlation analysis revealed a positive correlation between the number of glomeruli and that of antennal-expressed *AcerORs*. Specifically, queens and workers had much more glomeruli and antennal-expressing *AcerORs* than drones (Figure 9A). Although the number of glomeruli was slightly less than the number of *AcerORs* in drones, a nearly 1:1 ratio between glomeruli and *AcerORs* was seen in all castes. Interestingly, the *AcerORs* expression distribution and the glomerulus volume distribution were quite similar in all three castes. *AcerORs* expression level and the volume of glomerulus formed a pear-shaped curve, suggesting that most glomeruli are small in size, while a few are extremely large. The same trend was seen for *AcerORs* expression, i.e., most *AcerORs* expressions were low, but a few *AcerORs* were expressed extremely high (Figure 9B–D).

## 3. Discussion

### 3.1. Caste Dimorphism Association with AcerORs

In total, 106 *AcerORs* were identified in RNA-seq data, which is slightly lower than the 119 *AcerORs* in the genome [16]. Among most Hymenopteran species with available genome sequences (122–392 ORs), *A. cerana* had the lowest number of ORs [6,24,25]. Compared with its relative *A. mellifera*, the genome size and the number of genes in *A. cerana* are quite similar, but the number of ORs in *A. cerana* is much less [16]. It could be that the quality of genomic data from *A. cerana* is not as good as *A. mellifera* (Contig N50: 3.9 Mb) [26]. Gene expression profile analysis revealed that most *AcerORs* had antennal-biased expression; however, a few *AcerORs* were specifically expressed in non-olfactory organs, indicating their role in other biological processes rather than olfactory sensing [27,28]. The 9-ODA receptors (AcerOR11 and AmelOR11) are the 1:1 homology gene (97.46% similarity) between *A. mellifera* and *A. cerana*, indicating that sex pheromones sensing in these two species might be quite similar. Besides, some highly conserved ORs were detected in *A. cerana*, e.g., AcerOR151, with a 76.94% similarity to AmelOR151, which was shown to be activated by linalool [29].

In *A. cerana*, drones have fewer ORs than workers and queens, and similar trends could also be observed in other eusocial insects. For instance, only half of the ORs are expressed in males compared with workers in *Camponotus floridanus* and *Harpegnathos saltator* [11]. In eusocial insects, workers are responsible for more complex tasks, such as nectar collecting, hives cleaning, feeding queens, and so on. In comparison, drones only focus on reproduction, so this might be the cause of fewer ORs in drones. The antenna-specific ORs in drones such as *AcerOR11*, *AcerOR12*, *AcerOR18*, and *AcerOR22*, etc., possibly evolved for pheromones sensing, which can be examined in a future study.

### 3.2. Caste Dimorphism Association with Als

Both *A. cerana* and *A. mellifera* are cavity-nesting honeybees and are quite similar in behavior and physiology. However, the volume of their Als is quite different (*A. mellifera* workers: 10 × 10^6^ μm^3^; *A. cerana* workers: 3.34 × 10^6^ μm^3^) as well as the number of glomeruli (*A. mellifera* queens: ~150, drones: ~160, workers: ~104; *A. cerana* queens: ~116, drones: ~68, workers: ~117) [30]. These differences could be simply because *A. mellifera* had a larger body than that of *A. cerana*, or can be a reflection of the narrowing of the olfactory sensing spectrum in *A. cerana* [16].

The number of glomeruli was almost the same between queens and workers in *A. cerana*; however, the total volume of ALs in queens significantly decreased. A previous study showed that the development period of glomeruli in queens is shorter than in workers, which is possibly due to nutrient differences. This difference influences hormone secretion, gene expression, and neuropeptide synthesis, partially explaining the reason for the total volume decrease in ALs in queens [31,32].

In *A. cerana*, compared with females, males have fewer glomeruli (40% decreased); in other eusocial insects, this decrease is 20–60% [12,33,34]. The decrease in the number of glomeruli is consistent with the expression of ORs due to the narrowed olfactory sensing spectrum, which is enough for drones involved only in mating tasks.

When comparing the ALs structure of drones between *A. mellifera* and *A. cerana*, we found that the relative position of homology MGs was nearly the same, but their relative volume differs significantly. There are four MGs in the ALs of *A. mellifera* drones, named MGa, MGb, MGc, and MGd. Notably, MGa, MGb, and MGd homologous glomeruli exist in *A. cerana* drones, while the volume of MGc decreased. This reduced-volume glomerulus cannot be defined as an MG in *A. cerana* and is thus named Gc according to the classification rules [15]. Calcium imaging showed that the largest MGb of *A. mellifera* specifically responds to 9-ODA; however, other queen retinue pheromones (QRPs) components only activated the ordinary glomeruli [35]. The MGb in *A. cerana* was also the largest MG in its ALs, suggesting its role in QRPs sensing.

### 3.3. Correlation between AcerORs and Glomeruli

There are about 115 glomeruli in both queens and workers of *A. cerana*, and this number is roughly the same (slightly higher) as the number of ORs expressed in their antennae, indicating the “one neuron–one receptor–one glomerulus” law. Most insects meet this criterion, and several pieces of evidence verified this hypothesis, especially in the model insect *Drosophila melanogaster* [36]. However, sometimes two ORs are expressed in the same ORN [37], and ALs might also participate in transduction signals of ionotropic receptors (IRs) and gustatory receptors (GRs) [38]. This explains the slightly higher number of glomeruli compared with the number of *AcerORs* in *A. cerana*. There are other exceptions too, for example, *Culex quinquefasciatus* has 179 *ORs*, but only 62 and 44 glomeruli in females and males, respectively. One possibility is that some ORs are located very close in the chromosome and they project to the same glomerulus [39]. Another possibility is that several ORs are expressed in the same ORN or simply because some ORs are pseudogenes. Conversely, locusts have thousands of glomeruli, much more than ORs, which suggests that a single ORN can project onto multiple glomeruli [40,41].

The expression levels of ORs and the volume of glomeruli both showed a pear-shaped distribution, suggesting a positive correlation between them. There are three highly expressed *AcerORs* (FPKM value eight times higher than the average) and three MGs in drones, which possibly have a strict one-to-one correlation. In *A. mellifera* drones, *AmelOR11* and MGb specifically respond to 9-ODA. The homologous gene and homologous glomerular of *AmelOR11* and MGb also exist in *A. cerana* drones (*AcerOR11* and MGb), suggesting a similar 9-ODA sensing olfactory pathway between *A. cerana* and *A. mellifera* [42]. *AcerOR11* is the most highly abundant OR in drones, and MGb is also the biggest glomerulus in the ALs; a larger glomerulus indicates a larger number of ORNs [39]. This indicates that the 9-ODA pathway functions from *AcerOR11* to MGb from the peripheral to the CNS level. Meanwhile, these results also indicate the crucial role of 9-ODA sensing in *A. cerana* in enhancing the action of pheromones. Aside from 9-ODA, many other honeybee pheromones were identified; however, the olfactory sensing mechanism of these pheromones remains unclear. The biased expression of ORs and MGs in drones, identified in this study, possibly participates in pheromone sensing, which might improve our understanding of the complex pheromone communications in honeybees.

## 4. Materials and Methods

### 4.1. Insects

*A. cerana* bees used in this study were provided by the Jilin Provincial Institute of Apicultural Sciences, China. The bee population (northeast ecotype) was collected from Jilin, China (43°43′16″ N, 125°40′7″ E); the colony was reared since 2017. Queens (Q), drones (D), and workers (W) were cultured in an artificial incubator (Boxun, Shanghai, China) at 28 °C, relative humidity 70%, and a 16:8 light/dark cycle. Adult bees were fed with a 10% sucrose solution.

### 4.2. RNA Sequencing and Differential Gene Expression Analysis

RNA samples from different tissues, including antenna (A), proboscis (P), thorax (T), abdomen (Ab), and legs (L), were collected from 15 bee individuals per caste. The tissues were carefully isolated with DEPC (diethyl decarbonate)-treated forceps, cleaned with RNase-free water (Sangon Bio, Shanghai, China), and then temporarily stored on ice. Finally, they were immersed in liquid nitrogen for 1 h before storage at −80 ℃ for subsequent RNA extraction.

Total RNA was extracted using the TRIzol reagent (Invitrogen, Carlsbad, CA, USA) following the manufacturer’s protocols; the extracted RNA was subjected to ethanol/isopropanol precipitation. The concentration and purity of RNA samples were measured by NanoDrop 2000 spectrophotometer (Thermo Fisher Scientific, Waltham, MA, USA). The integrity of RNA was checked by 1% agarose gel electrophoresis, Qubit 2.0 Fluorometer (Life Technologies, Grand Island, NE, USA), and Bioanalyzer 2100 (Agilent, Palo Alto, CA, USA). The qualified RNA was used to construct cDNA libraries that were sequenced on a HiSeq 2000 PE150 platform (Illumina, New England Biolabs, Ipswich, MA, USA).

Raw reads were processed with in-house Perl scripts to remove the adapters, reads containing poly-N and low quality reads to obtain clean reads. At the same time, Q20, Q30, GC-content, and sequence duplication levels of the clean data were calculated. The clean reads were subsequently mapped to the reference genome of *A. cerana* (Accession No.: ApisCC1.0). Hisat2 tools were used for reference genome mapping, and StringTie was applied to assemble the mapped reads. Clean reads were annotated using the following databases: Nr (NCBI non-redundant protein sequences), Nt (NCBI non-redundant nucleotide sequences), Pfam (Protein family), KOG/COG (Clusters of Orthologous Groups of proteins), Swiss-Prot (A manually annotated and reviewed protein sequence database), KO (KEGG Ortholog database), and GO (Gene Ontology).

DESeq2 v 1.30.1 software was used to compare differentially expressed genes (DEGs) among different castes in *A. cerana*. The resulting P values were adjusted using the Benjamini and Hochberg approach for controlling the false discovery rate. Genes with an adjusted *p*-value < 0.01 and Fold Change ≥ 2 found by DESeq2 were assigned as differentially expressed. The obtained DEGs were further annotated by Gene Ontology (GO).

Candidate *AcerORs* obtained from annotation results were blasted against the NCBI non-redundant protein sequences database (nr, https://blast.ncbi.nlm.nih.gov/Blast.cgi) (accessed on 21 September 2022) to acquire annotation results. Subsequently, an *AcerORs* dataset was established to perform a second round of local blasts to dig out new *AcerORs* genes. Open reading frames (ORFs) were identified using the ORF finder (https://www.ncbi.nlm.nih.gov/orffinder/) (accessed on 21 September 2022), most ORs had 7 transmembrane domains (TMDs), and the TMDs were predicted by TMHMM server version 2.0 (http://www.cbs. Dtu.dk/services/TMHMM/) (accessed on 21 September 2022) with default parameters.

### 4.3. Sequence Homology and Expression Pattern Analysis of AcerORs

For homology analysis, ORs from *A. mellifera* and *A. florea* were also picked [10,43], and after the ClustalW alignment of sequences, a maximum likelihood (ML) tree was constructed by Jones–Taylor–Thornton (JTT) model in MEGA X with a bootstrap value of 1000. The *AcerORs* were named by their homological relationship to *A. mellifera*. The ML tree was visualized by FigTree v1.4.3, and Fragments Per Kilobase of transcript per Million fragments mapped (FPKM) were used to measure the gene expression. The heatmap of gene expression data was constructed in R v4.1.3.

### 4.4. Real-Time Quantitative PCR (RT-qPCR) Validation of RNA-Seq Data

Selected *AcerORs* DEGs in the antenna of different castes were validated by RT-qPCR. Corresponding cDNAs were synthesized with the First-Strand cDNA Synthesis Kit (Transgen Biotech, Beijing, China) using 1 μg of total RNA. cDNA was diluted 10-fold for use as the template in RT-qPCR. The *β-actin* gene was used as the reference gene to normalize the expression data. Specific primers were designed using Primer 3 (https://bioinfo.ut.ee/primer3-0.4.0/) (accessed on 8 June 2023) and are listed in Appendix A. RT-qPCR was conducted on the LightCycler 480 II Detection System (Roche, Basel, Switzerland) and TransStar Tip Top Green qPCR Supermix (Transgen Biotech, Beijing, China). RT-qPCR was set up according to the manufacturer’s instructions, with the following conditions: 94 °C for 30 s, followed by 45 cycles of 94 °C for 5 s, 55 °C for 15 s, and 72 °C for 10 s. The gene expressions were calculated using the 2^−ΔΔCT^ method. All qPCRs were conducted with three technical and three biological replicates.

### 4.5. Immunohistochemical Staining

The bee brain was carefully separated by forceps and fixed with 4% paraformaldehyde fixing solution (PFA) at 4 °C overnight. The next day, the brain was rinsed with PBST (PBS containing 0.5% Triton X-100; 0.01M of PBS buffer: 1.8 mM of KH_2_PO_4_, 4.3 mM of Na_2_HPO_4_, 137 mM of NaCl, 2.7 mM of KCl, pH = 7.4), 15 min each time. The fixed brain was then preincubated with 5% normal goat serum (NGS; Thermo Fisher Scientific, Waltham, MA, USA) at 4 °C for 15 h. Afterward, the waste solution was sucked out, and the brain was incubated with 3C11 primary antibodies (anti SYNORF1, 1:100 dilution in 5% NGS containing PBST) (DSHB, University of Iowa, Iowa City, IA, USA) at 4 °C for 5 days. After that, the brain was rinsed with PBST 6 times, followed by incubation with Cy2-coupled Alexa Fluor^TM^ 488 secondary antibodies (1:300 dilution in 1% NGS containing PBST) (Invitrogen, Eugene, OR, USA) at 4 °C for 3 days. Finally, the brain was rinsed 6 times with PBST to remove unbound antibodies, dehydrated with gradient alcohol, cleared with methyl salicylate, and then stored at 4 °C.

### 4.6. Confocal Laser Microscopy

The respective brain sample was carefully mounted in a circular hole of a 1 mm aluminum slide. A small amount of neutral balsam (Solarbio, Beijing, China) was added to seal the slide with coverslips, and the bubbles were carefully removed to avoid interferences during imaging. Slides were observed under a laser scanning confocal microscope (LSM880, Carl Zeiss, Jena, Germany) at an excitation wavelength of 488 nm, and images were collected between 490 and 560 nm. The microscope parameters were as follows: unidirectional scanning averaging line, 2; dimension, 1024 × 1024; scanning speed, 6 or 7; scanning layers, 2 or 3 μm. Clear images were captured by ZEN v2.6.

### 4.7. 3D model Reconstruction

Previous studies reported that the location, volume, and shape of glomeruli significantly differ based on bee caste, which makes it difficult to identify homologous glomeruli [32]. Therefore, 3D models were reconstructed independently in different castes. The confocal image stacks were imported into 3D reconstruction software Amira v5.4.3, and the glomeruli were named based on their location (A, anterior; P, posterior; D, dorsal; V, ventral; L, lateral; M, medial; C, central). The volume and surface area of each glomerulus were estimated by “Label Analysis” in the software, and the types were classified based on glomeruli shape and location. The confocal images that were not clear enough for 3D reconstruction were used to calculate the number of glomeruli by Image J v1.8.0. In addition, the shape of glomeruli was defined by their volume and surface area [44].

### 4.8. Statistics

The significant difference of gene expression, qPCR, and 3D modelling data were calculated by one-way ANOVA followed by Tukey’s multiple comparison test (*p* < 0.05) after checking the normality and homogeneity of variance. GraphPad Prism v8.0 was used to generate all graphs.

## 5. Conclusions

This study shows that the expression of *AcerORs* and the architecture of glomeruli are associated with caste dimorphism in *A. cerana*. The numbers of *AcerORs* and glomeruli and the volume of glomeruli were decreased in drones, indicating their specialization in reproduction-related tasks. *AcerORs* expression profiles and the number of glomeruli were similar between queens and workers, but the total volume and the surface area of ALs were significantly reduced in queens, indicating developmental heterochrony. Furthermore, *AcerORs* expressed in the antenna and the number of glomeruli in ALs exhibited a ratio of 1:1; meanwhile, the *ORs* expressing level and the glomeruli volume exhibited a similar distribution pattern. Concisely, this study comprehensively improves our understanding of the olfactory differentiation based on caste dimorphism in eusocial insects, which can benefit the management, protection, and conservation of natural pollinators.

## Figures and Tables

**Figure 1 ijms-25-03934-f001:**
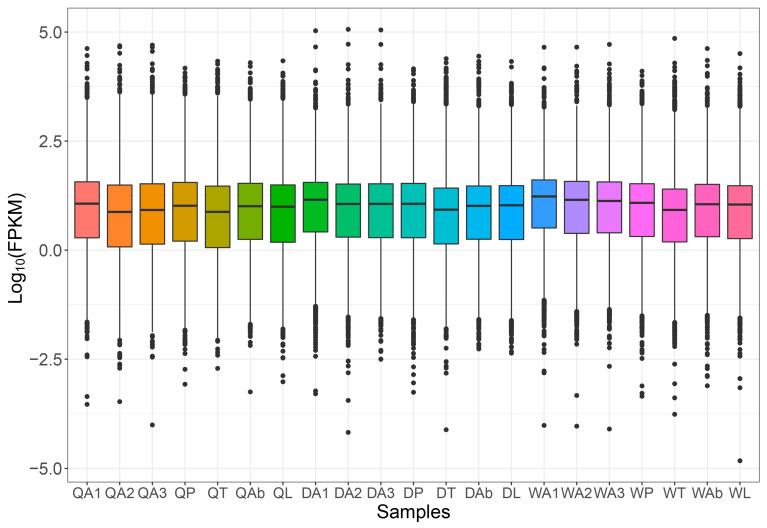
A box plot showing FPKM values in 21 tissues of different castes of *A. cerana*. QA: Queen antenna; QP: Queen proboscis; QT: Queen thorax; QAb: Queen abdomen; QL: Queen legs; DA: Drone antenna; DP: Drone proboscis; DT: Drone thorax; DAb: Drone abdomen; DL: Drone legs; WA: Worker antenna; WP: Worker proboscis; WT: Worker thorax; WAb: Worker abdomen; WL: Worker legs.

**Figure 2 ijms-25-03934-f002:**
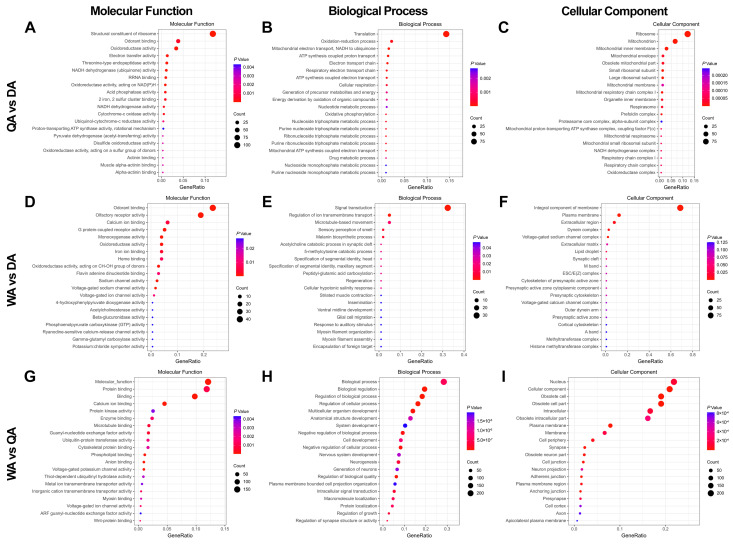
The unigenes GO enrichment analysis of upregulated genes in the antenna of different castes of *A. cerana*. (**A**–**C**) QA vs. DA group; (**D**–**F**) WA vs. DA group; (**G**–**I**) WA vs. QA group. The ordinate shows the information of the top 20 GO nodes with the *p*-value. The abscissa represents the “GeneRatio”, the total proportion of genes related to the corresponding GO node. The size and color of the circle denote the count and *p*-value of the classification, respectively.

**Figure 3 ijms-25-03934-f003:**
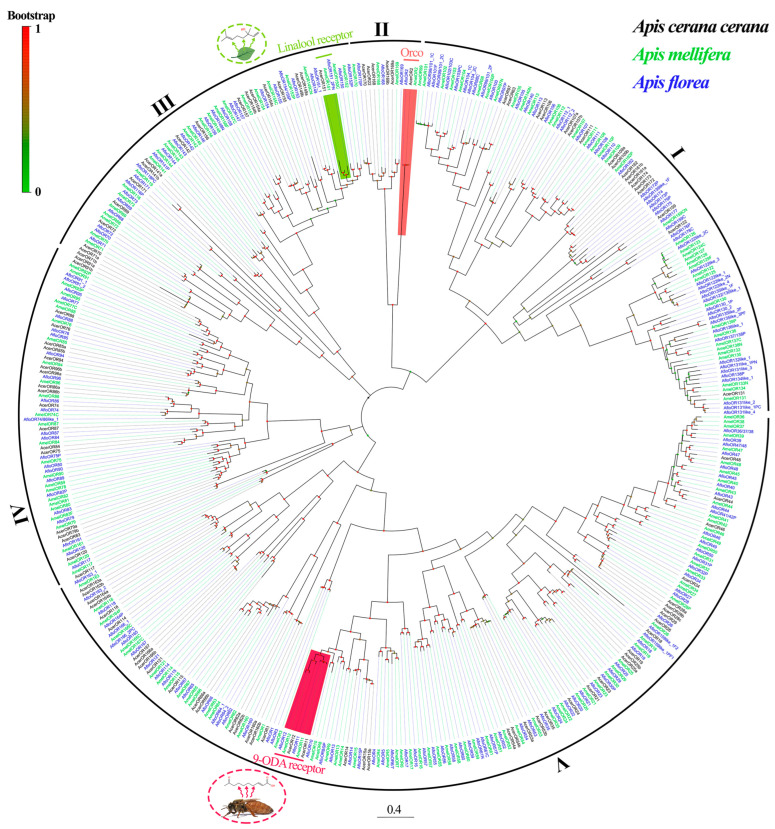
The maximum likelihood tree of Orco and ORs: AcerORs (black), AmelORs (green), and AfloORs (blue). Bootstrap replications up to 1000. The lowercase alphabets after the protein name indicate alternative splicing.

**Figure 4 ijms-25-03934-f004:**
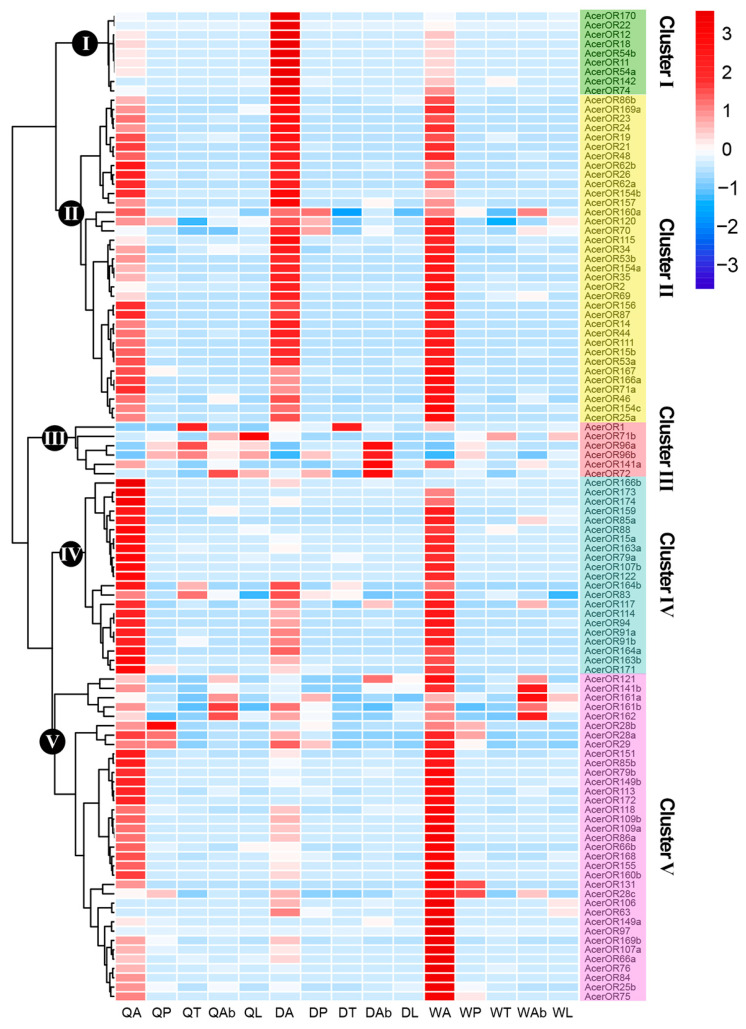
Expression profiles of *AcerORs* in *A. cerana.* Five clusters denote the characteristic expression patterns of *AcerORs* in different tissues based on FPKM (normalization by row). The clustering method was ward.D2. FPKM: Fragments Per Kilobase of transcript per Million fragments mapped.

**Figure 5 ijms-25-03934-f005:**
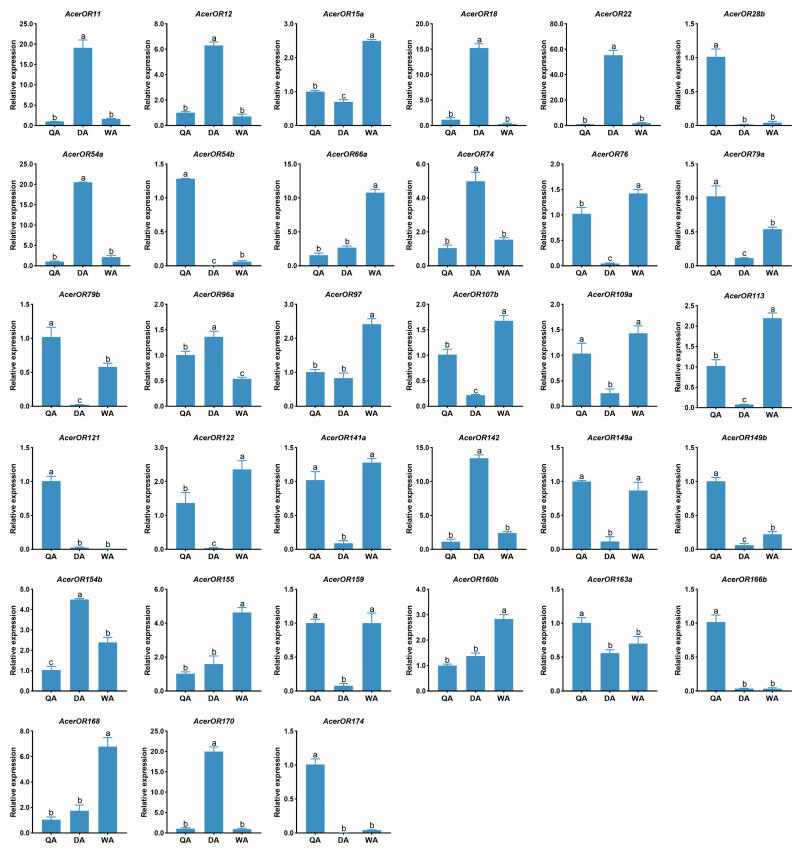
RT-qPCR results of 33 candidate *AcerORs* in the antennae of different castes in *A. cerana*. QA: Queen antenna; DA: Drone antenna; WA: Worker antenna. Different lower case letters indicates significant differences based on a one-way ANOVA followed by Tukey’s multiple comparison test (*p* < 0.05). Data indicate the mean ± standard error of the mean (SEM).

**Figure 6 ijms-25-03934-f006:**
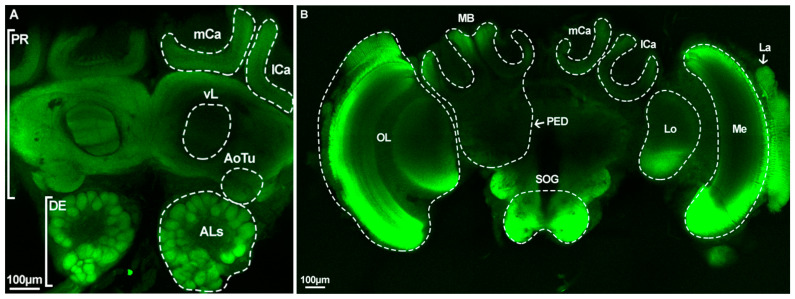
Overall brain structure of *A. cerana* worker. (**A**) ventral perspective; (**B**) dorsal perspective. PR: protocerebrum; DE: deutocerebrum; MB: mushroom body; OL: optic lobe; PED: pedunculus; mCa: medial calyx; lCa: lateral calyx; vL: vertical lobe; AoTu: anterior optic tubercle; ALs: Antennal lobes; SOG: subesophageal ganglion; Lo: lobula; Me: medulla; La: lamina.

**Figure 7 ijms-25-03934-f007:**
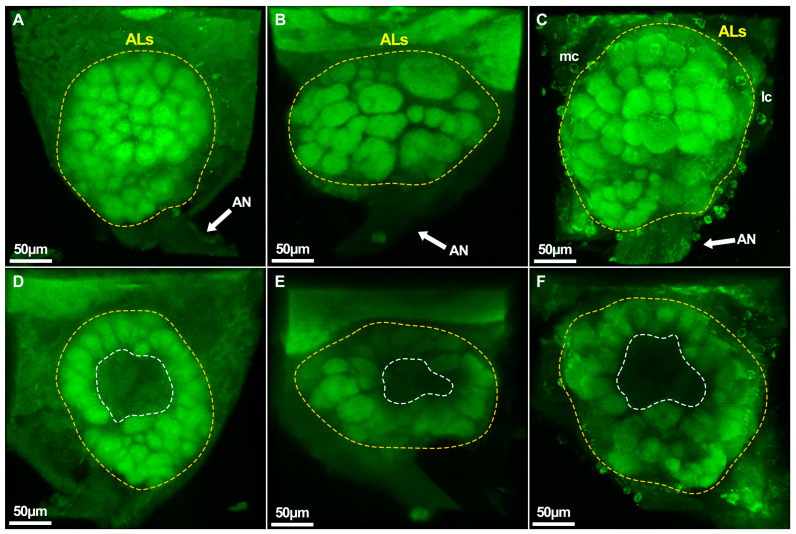
The appearance of *A. cerana* antennal lobes. (**A**–**C**) ventral perspective; (**D**–**F**) dorsal perspective; antennal lobes of (**A**,**D**) queen, (**B**,**E**) drone, and (**C**,**F**) worker. ALs: Antennal lobes; AN: Antennal nerve; mc: medial cluster of ALs neuron somata; lc: lateral cluster of antennal lobe neuron somata. The dashed yellow areas are ALs and dashed white areas indicate the absence of glomeruli.

**Figure 8 ijms-25-03934-f008:**
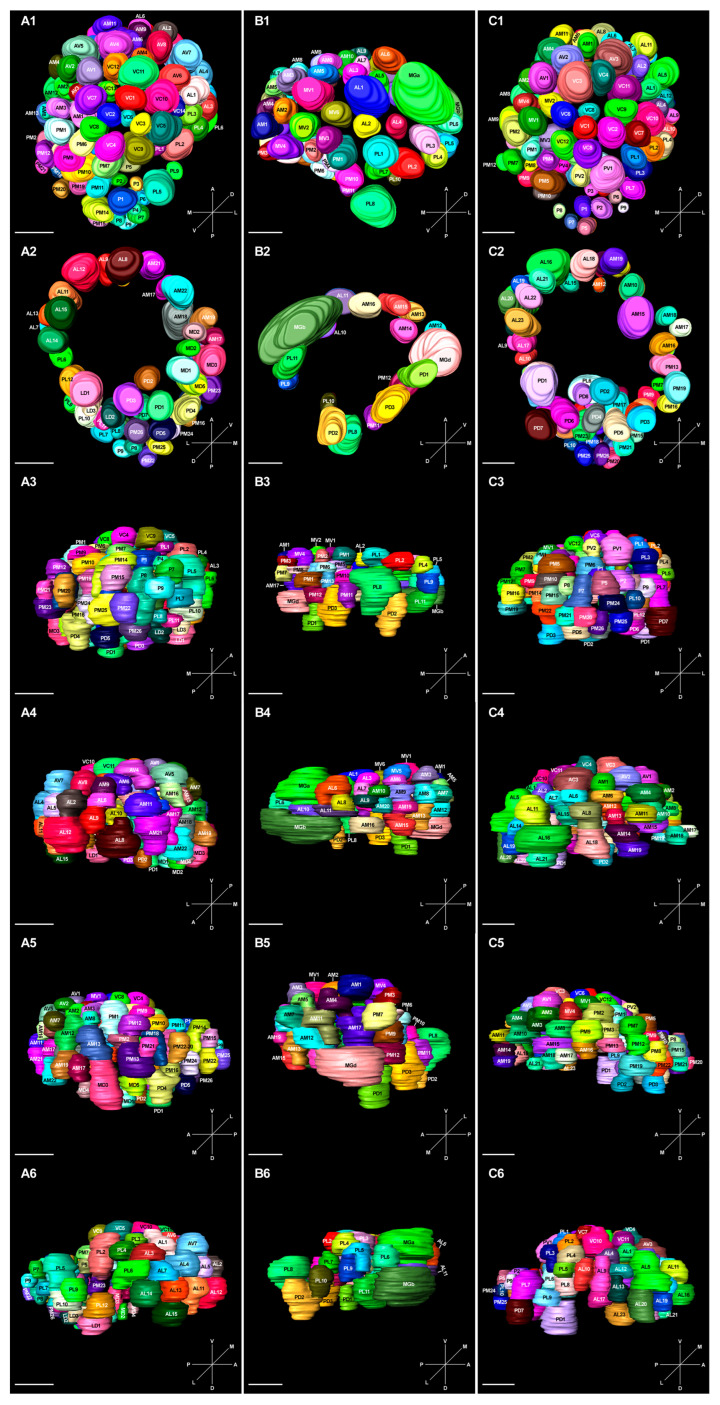
Three-dimensional reconstruction of left antennal lobes (ALs) of *A. cerana*. ALs of (**A1**–**A6**) queen, (**B1**–**B6**) drone, and (**C1**–**C6**) worker are shown. (**A1**,**B1**,**C1**): Ventral view; (**A2**,**B2**,**C2**): Dorsal view; (**A3**,**B3**,**C3**): Posterior view; (**A4**,**B4**,**C4**): Anterior view; (**A5**,**B5**,**C5**): Medial view; (**A6**,**B6**,**C6**): Lateral view. Scale bar = 50 μm.

**Figure 9 ijms-25-03934-f009:**
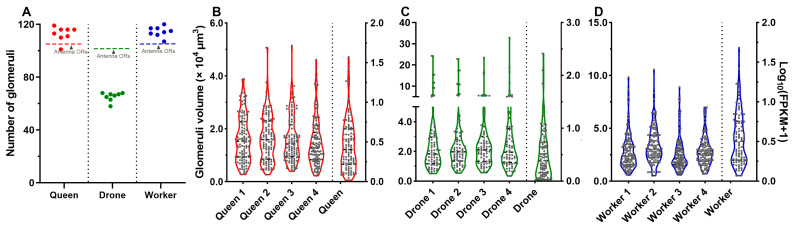
Correlation analysis between glomeruli and *AcerORs*. (**A**) Relationship between the number of glomeruli (*n* = 8) and antennal-expressed *AcerORs*. Solid circles and the horizontal dotted lines indicate the number of glomeruli and that of antennal-expressed *AcerORs*, respectively. Data indicate the mean ± standard error of the mean (SEM). (**B**–**D**) Distribution of certain glomerular volume and *AcerORs* expression levels.

**Table 1 ijms-25-03934-t001:** The genes of DEGs in different castes of *A. cerana*: QA: Queen antenna; DA: Drone antenna; WA: Worker antenna.

Groups	Total DEGs	Upregulated Genes	Downregulated Genes
QA vs. DA	3252	1594	1658
WA vs. DA	568	317	251
WA vs. QA	3866	2100	1766

**Table 2 ijms-25-03934-t002:** *AcerORs* number expression in different tissues (FPKM = 0 denotes that the gene was not expressed).

Caste	Total	Antenna	Proboscis	Thorax	Abdomen	Legs
Queen	106	106	46	40	60	58
Drone	101	99	46	44	58	41
Worker	106	106	55	36	63	63

**Table 3 ijms-25-03934-t003:** Characteristics of antennal lobes and the number of glomeruli types. Different letters indicate significant differences between groups (*p* < 0.05). Data indicate the mean ± standard error of the mean (SEM).

	Queens	Drones	Workers
Total number of glomeruli	116 ± 2.9 a	68 ± 0.5 b	117 ± 2.3 a
Total volume of ALs (×106 μm^3^)	1.82 ± 0.06 b	2.04 ± 0.14 b	3.34 ± 0.35 a
Total surface area of ALs (×105 μm^2^)	4.01 ± 0.09 b	3.74 ± 0.08 b	6.07 ± 0.43 a
AL type	15	11	24
AM type	22	20	19
AV type	8	-	3
MV type	-	7	4
LD type	3	-	-
MD type	5	-	-
PD type	7	3	8
P type	9	-	9
PL type	14	16	17
PM type	28	14	26
PV type	-	-	4
MG/G type	-	4	-

## Data Availability

The datasets analyzed in the current study are available from the corresponding author upon reasonable request.

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
