# Peer review of "Odorant Receptors Expressing and Antennal Lobes Architecture Are Linked to Caste Dimorphism in Asian Honeybee, Apis cerana (Hymenoptera: Apidae)"

_ijms, 2024, doi:10.3390/ijms25073934_

Round 1

Reviewer 1 Report (Previous Reviewer 3)

Comments and Suggestions for Authors

The manuscript has significant value in science. However, some sections need to be rewritten compactly. A few sections (3D model of glomeruli) are not very informative. Therefore, vigorous revision is needed.

Line 12: Add text about the aims of the study after the research knowledge gap.

Line 22-23: Keywords:--- arrange alphabetically, avoid plural forms.

Line 39-44: For which honeybee species these statements are given? Need to rewrite more clearly. Another issue is that the references 11-13 came from other insects rather than honeybees.

Line 47-49: Why they compared Apis cerana with Apis mellifera only?

Line 56-57: high pollination efficiency is an irreplaceable pollinator---- Rewrite. What do you mean by irreplaceable?

Line 57-59: Add references like Pettis et al. 2012 Naturwissenschaften doi: 10.1007/s00114-011-0881-1; Layek et al. 2021 Annals of Agricultural Sciences 66: 38-45. doi: 10.1016/j.aoas.2021.02.004; Insolia et al. 2022 Scientific Reports 12, 20787. doi: 10.1038/s41598-022-24946-4

Line 60: Add text to explain the necessity of the research work and the research gap.

Line 71: Table. S2---- Follow the journal's style. Check throughout the manuscript. Also, consider sequential numbering patterns for Tables and Figures.

Line 74-75: one-way ANOVA...... comparison test---- delete this part. Add F value and degree of freedom.

Line 111-116: This part seems to be more likely Discussion. Add data sources used here for other insect species.

Line 176-177: This sentence is not fitting here.

Line 184-193: This paragraph is just repeating the above paragraph. Therefore, need to rewrite these two paragraph compactly.

Line 238-239: Most Hymenoptera species....... [21]---- add more references.

Line 310: lager----check.

Line 418-420: Use the subheading 'Statistical analyses' and detailed statistical analysis needed to be given.

References list---- Check the journal's name in proper abbreviation (e.g., line 470).

Author Response

Response to Reviewer 1 Comments

The manuscript has significant value in science. However, some sections need to be rewritten compactly. A few sections (3D model of glomeruli) are not very informative. Therefore, vigorous revision is needed.

[Response]: We feel great thanks for your professional review work on our article. Our detailed responses are as following:

Line 12: Add text about the aims of the study after the research knowledge gap.

[Response]: Thank you for the suggestion. We have added study aims in abstract.

Line 13-15: To explore the peripheral and primary center of olfactory system link to the caste dimorphism in A. cerana, transcriptome and immunohistochemistry studies on the odorant receptors (ORs) and architecture of antennal lobes (ALs) were performed on different castes.

Line 22-23: Keywords:--- arrange alphabetically, avoid plural forms.

[Response]: Thank you, the keywords were re-ordered.

Line 25-26: Keywords: Apis olfactory system; antennal lobes; bee glomeruli; caste dimor-phism; odorant receptor

Line 39-44: For which honeybee species these statements are given? Need to rewrite more clearly. Another issue is that the references 11-13 came from other insects rather than honeybees.

[Response]: Thank you for the carefully check. To be more rigorous, we have corrected “In honeybees” into “In eusocial insects”.

Line 40: In eusocial insects, caste dimorphism of the olfactory system…

Line 47-49: Why they compared Apis cerana with Apis mellifera only?

[Response]: In the genus Apis, A. mellifera and A. cerana are both cavity-nesting bees, they were most close-related species, whatever in genome or morphology. Moreover, as a model species, a large of studies are focusing on A. mellifera which provide an informative reference to our studies.

Lin, D.; Lan, L.; Zheng, T.; Shi, P.; Xu, J.; Li, J. Comparative Genomics Reveals Recent Adaptive Evolution in Himalayan Giant Honeybee Apis laboriosa. Genome Biol. Evol. 2021, 13, evab227.

Line 56-57: high pollination efficiency is an irreplaceable pollinator---- Rewrite. What do you mean by irreplaceable?

[Response]: Thank you for the check, the words “is an irreplaceable pollinator” was removed.

Line 59-63: A. cerana with a wide range of host plants and high pollination efficiency, however, the olfactory system of A. cerana is poorly understood compared with its close relative A. mellifera. The colonies of A. cerana are on a significant decline due to the massive use of pesticides and pathogens invasion, especially with the introduction of A. mellifera in commercial apiculture.

Line 57-59: Add references like Pettis et al. 2012 Naturwissenschaften doi: 10.1007/s00114-011-0881-1; Layek et al. 2021 Annals of Agricultural Sciences 66: 38-45. doi: 10.1016/j.aoas.2021.02.004; Insolia et al. 2022 Scientific Reports 12, 20787. doi: 10.1038/s41598-022-24946-4

[Response]: We have added these references to support this idea.

Line 60: Add text to explain the necessity of the research work and the research gap.

[Response]: Thank you for the comments, the following sentences was added in Line 59-63.

Line 59-63: A. cerana with a wide range of host plants and high pollination efficiency, however, the olfactory system of A. cerana is poorly understood compared with its close relative A. mellifera. The colonies of A. cerana are on a significant decline due to the massive use of pesticides and pathogens invasion, especially with the introduction of A. mellifera in commercial apiculture.

Line 71: Table. S2---- Follow the journal's style. Check throughout the manuscript. Also, consider sequential numbering patterns for Tables and Figures.

[Response]: Thank you for your carefully check. The format were corrected throughout the manuscript.

Line 74-75: one-way ANOVA...... comparison test---- delete this part. Add F value and degree of freedom.

[Response]: The sentence “one-way ANOVA followed by Tukey’s multiple comparison test” was removed and the F value and df were added in the manuscript.

Line 76-78: Among different castes, the average gene expression levels in the antenna of queens were significantly higher than that of drones and workers (F = 16.181, df = 2, p < 0.05, Figure 1, Table S3).

Line 111-116: This part seems to be more likely Discussion. Add data sources used here for other insect species.

[Response]: Thank you for the correction. The text was rewritten as follows:

Line 114-119: The best hit of most AcerORs were the orthologues ORs from A. mellifera, while the some of AcerORs (such as AcerOR34, AcerOR85b, and AcerOR91b etc.) were best hit to the ORs in Apis dorsata, Apis laboriosa, and Apis florea. In addition, some AcerORs (AcerOR97, AcerOR155, and AcerOR159 etc.) were also found to be most similar to the ORs of Bombus vancouverensis nearcticus, Bombus terrestris, and Temnothorax curvispinosus (Table S5).

Line 176-177: This sentence is not fitting here.

[Response]: The sentence was deleted.

Line 184-193: This paragraph is just repeating the above paragraph. Therefore, need to rewrite these two paragraph compactly.

[Response]: Thank you for the correction, the relevant text was moved to the above paragraph and compacted.

Line 177-184: We found that although the shape of ALs varied among castes, the distribution of glomeruli was quite similar. The ALs of queens and workers were almost round in shape but those of drones exhibited an oval shape. Queens and workers have several small glomeruli at the entrance of the antennal nerve to the ALs but the same was re-placed by several larger glomeruli in drones. In all three castes, glomeruli were scattered at the edge of ALs, forming a hollow central fiber core without synaptic. In general, there were more ventral side glomeruli than on the dorsal side. Most glomeruli were nearly spherical and ellipsoidal but a few glomeruli were irregular in shape (Figure 7, S1).

Line 192-196: The 3D modelling results showed the number of glomeruli increases from the pos-terior to the anterior side. A few large glomeruli were observed in the middle of the outermost portion of the ventral side but no glomeruli were found on the opposite side. All the glomeruli formed a hollow sphere and more glomeruli were distributed in the equatorial axis than the pole sides located in the outermost portion of the sphere. Moreover, there were several MGs in drones that were located at the edge of ALs (Fig-ure 8, S2).

Line 238-239: Most Hymenoptera species....... [21]---- add more references.

[Response]: We have added more references to support this idea.

Line 241-242: Among most Hymenopteran species with available genome sequences (122 - 392 ORs), A. cerana had the lowest number of ORs [6,24,25].

  1. Zhou, X.; Rokas, A.; Berger, S.L.; Liebig, J.; Ray, A.; Zwiebel, L.J. Chemoreceptor Evolution in Hymenoptera and Its Implications for the Evolution of Eusociality. Genome Biol. Evol. 2015, 7, 2407-2416, doi:10.1093/gbe/evv149.
  2. Obiero, G.F.; Pauli, T.; Geuverink, E.; Veenendaal, R.; Niehuis, O.; Grosse-Wilde, E. Chemoreceptor Diversity in Apoid Wasps and Its Reduction during the Evolution of the Pollen-Collecting Lifestyle of Bees (Hymenoptera: Apoidea). Genome Biol. Evol. 2021, 13, evaa269, doi:10.1093/gbe/evaa269.
  3. Karpe, S.D.; Dhingra, S.; Brockmann, A.; Sowdhamini, R. Computational genome-wide survey of odorant receptors from two solitary bees Dufourea novaeangliae (Hymenoptera: Halictidae) and Habropoda laboriosa (Hymenoptera: Apidae). Sci. Rep. 2017, 7, 10823.

Line 310: lager----check.

[Response]: Corrected.

Line 418-420: Use the subheading 'Statistical analyses' and detailed statistical analysis needed to be given.

[Response]: An additional subheading was added in the relevant place.

Line 425-428: The significant difference of gene expression, qPCR and 3D modelling data were calculated by one-way ANOVA followed by Tukey’s multiple comparison test (p < 0.05) after checked the normality and homogeneity of variance. GraphPad Prism v8.0 was used to generated all graphs.

References list---- Check the journal's name in proper abbreviation (e.g., line 470).

[Response]: Thank you for your carefully check, they were corrected throughout the references list.

Reviewer 2 Report (New Reviewer)

Comments and Suggestions for Authors

The topic of the article is very interesting. Knowing the composition of odor receptors in bees allows us to understand their physiology as well.

There are two female castes in bees: the worker bee and the queen bee. The queen bee and the worker bee are born from fertilized eggs with the same genetic set but undergo a change in diet during larval feeding that causes them to differentiate into the two castes. The drone does not represent a caste but arises from unfertilized and therefore haploid eggs. I ask the authors to correct the paper by not including the drones among the castes but possibly reporting the differences as sex difference.

The section on statistical analysis is missing in the materials and methods. What analysis was done? At one point in the text it mentions data tested for anova followed by Tukey's test, but were the data tested for normality and homogeneity? If not, you cannot apply anova but a nonparametric test.

"vs" should be written in italics. Please correct it.

Figures 2, 3, 4 and 5 do not read anything. Please enlarge them.

I suggest citing the following article as it was one of the first papers that is addressed with odor receptors on bees: Yin, X. W. Et al. (2013). Odorant-binding proteins and olfactory coding in the solitary bee Osmia cornuta. Cellular and molecular life sciences70, 3029-3039.

Comments on the Quality of English Language

Minor editing of English language required

Author Response

Response to Reviewer 2 Comments

The topic of the article is very interesting. Knowing the composition of odor receptors in bees allows us to understand their physiology as well.

There are two female castes in bees: the worker bee and the queen bee. The queen bee and the worker bee are born from fertilized eggs with the same genetic set but undergo a change in diet during larval feeding that causes them to differentiate into the two castes. The drone does not represent a caste but arises from unfertilized and therefore haploid eggs. I ask the authors to correct the paper by not including the drones among the castes but possibly reporting the differences as sex difference.

[Response]: Thank you for your valuable suggestions, indeed, as you mentioned, queens and workers are female while drones are males, and their olfactory system are similar in some aspects, however, in our study, we also found that there is obvious differentiation between queens and worker (both females) whatever in the olfactory genes or the architecture of their antennal lobes, meanwhile, queens and workers are highly diverse in their biology because queens are only responsible for mating and oviposition, as workers responsible for variety tasks such as pollination, foraging, brooding etc., they are facing a totally different chemical sensing environments.

Thus, as you suggested, we corrected those results that no big differences between queens and workers as a “sex difference”, meanwhile we defined those differences between queens and workers as a “caste difference” throughout the manuscript, thank you for the reminds.

The section on statistical analysis is missing in the materials and methods. What analysis was done? At one point in the text it mentions data tested for anova followed by Tukey's test, but were the data tested for normality and homogeneity? If not, you cannot apply anova but a nonparametric test.

[Response]: Thank you for your carefully check, we have added one more section about the statistical analysis.

Line 425-428: The significant difference of gene expression, qPCR and 3D modelling data were calculated by one-way ANOVA followed by Tukey’s multiple comparison test (p < 0.05) after checked the normality and homogeneity of variance. GraphPad Prism v8.0 was used to generated all graphs.

"vs" should be written in italics. Please correct it.

[Response]: Corrected, thank you.

Figures 2, 3, 4 and 5 do not read anything. Please enlarge them.

[Response]: Figure 2, 3, 4 and 5 were enlarged, and we will also upload the separated Figure file in the final version, thank you.

I suggest citing the following article as it was one of the first papers that is addressed with odor receptors on bees: Yin, X. W. Et al. (2013). Odorant-binding proteins and olfactory coding in the solitary bee Osmia cornuta. Cellular and molecular life sciences, 70, 3029-3039.

[Response]: The above article is cited, thank you for providing this valuable information, it is a fantastic work which could leading our future studies.

Round 2

Reviewer 1 Report (Previous Reviewer 3)

Comments and Suggestions for Authors

The authors well responded to the comments/suggestions. The revised version is more improved than the earlier version. Now, the manuscript is suitable for acceptance.

Reviewer 2 Report (New Reviewer)

Comments and Suggestions for Authors

I appreciate the changes the authors have made to the manuscript by improving it, however they continue to define the drones a caste, in all sections of the manuscript, while I understood that they had changed this definition. Did the authors perhaps upload an old version?

In the answers: I think you meant to write that the sex difference is between males and females not between workers and queens.

It is not a definition given by me but many authors do not call drones a caste, however I do not insist.

Engels, W., & Imperatriz-Fonseca, V. L. (1990). Caste development, reproductive strategies, and control of fertility in honey bees and stingless bees. In Social insects: an evolutionary approach to castes and reproduction (pp. 167-230). Berlin, Heidelberg: Springer Berlin Heidelberg.

Wojciechowski, M., Lowe, R., Maleszka, J., Conn, D., Maleszka, R., & Hurd, P. J. (2018). Phenotypically distinct female castes in honey bees are defined by alternative chromatin states during larval development. Genome research28(10), 1532-1542.

Peso, M., Even, N., Søvik, E., Naeger, N. L., Robinson, G. E., & Barron, A. B. (2016). Physiology of reproductive worker honey bees (Apis mellifera): insights for the development of the worker caste. Journal of Comparative Physiology A202, 147-158.

This manuscript is a resubmission of an earlier submission. The following is a list of the peer review reports and author responses from that submission.

Round 1

Reviewer 1 Report

Comments and Suggestions for Authors

N/A

Comments on the Quality of English Language

N/A

Reviewer 2 Report

Comments and Suggestions for Authors

Insects heavily rely on the olfactory system for food, mating, and predator evasion. However, the caste-related olfactory differences in Apis cerana cerana, a eusocial insect, remain unclear. This study found that the existence of concurrent plasticity in both the peripheral olfactory system and ALs among different castes of A. cerana, highlighting the role of the olfactory system in labor division in insects.

Here I suggest some minor suggestions.

1. The colony number of apis cerana is increasing in China, not decreasing. The authors should check out the latest data.

2. Discussion is not a reproduction of the results. The discussion part is too long but insufficient

Reviewer 3 Report

Comments and Suggestions for Authors

The work has significant value. After minor revision, the manuscript will be suitable for acceptance. My few comments are given below:

Line 22: keywords: "Apis olfactory system" change to "olfactory system; Apis cerana". Avoid plural forms. 

Line 47: Apis mellifera and A. cerana-- use the full name with author citation for the first time in the text.

line 50-51: ... collecting larger pollen form--- rewrite the sentence.

line 56-60: not good fitting here. This part may be more suitable in lines 49-51.

line 71,72: Table. S2, Table. S3--- Use the Table's name sequentially, start with Table S1 (which is in line 385). 

line 75-76: .... one way ANOVA followed by Tukey's multiple comparison test--- not necessary here. Statistical significance differences need to be shown in Figure 1. Tukey's test needs to be mentioned in the ligand of Figure 1.

Check the journal's abbreviation in the reference list (lines 475, 479, 531, 539, etc)
